# Myocardial NADPH oxidase-4 regulates the physiological response to acute exercise

Matthew Hancock[1], Anne D Hafstad[2], Adam A Nabeebaccus[1], Norman Catibog[1], Angela Logan[3], Ioannis Smyrnias[1], Synne S Hansen[2], Johanna Lanner[4], Katrin Schröder[5], Michael P Murphy[3], Ajay M Shah[1]*, Min Zhang[1]*

[1]School of Cardiovascular Medicine & Sciences, King's College London British Heart Foundation Centre of Excellence, London, United Kingdom; [2]Cardiovascular Research Group, Department of Medical Biology, University of Tromsø - The Arctic University of Norway, Tromsø, Norway; [3]MRC Mitochondrial Biology Unit, University of Cambridge, Cambridge, United Kingdom; [4]Department of Physiology and Pharmacology, Karolinska Institutet, Stockholm, Sweden; [5]Institut für Kardiovaskuläre Physiologien, Goethe-Universität, Frankfurt, Germany

**Abstract** Regular exercise has widespread health benefits. Fundamental to these beneficial effects is the ability of the heart to intermittently and substantially increase its performance without incurring damage, but the underlying homeostatic mechanisms are unclear. We identify the ROS-generating NADPH oxidase-4 (Nox4) as an essential regulator of exercise performance in mice. Myocardial Nox4 levels increase during acute exercise and trigger activation of the transcription factor Nrf2, with the induction of multiple endogenous antioxidants. Cardiomyocyte-specific Nox4-deficient (csNox4KO) mice display a loss of exercise-induced Nrf2 activation, cardiac oxidative stress and reduced exercise performance. Cardiomyocyte-specific Nrf2-deficient (csNrf2KO) mice exhibit similar compromised exercise capacity, with mitochondrial and cardiac dysfunction. Supplementation with an Nrf2 activator or a mitochondria-targeted antioxidant effectively restores cardiac performance and exercise capacity in csNox4KO and csNrf2KO mice respectively. The Nox4/Nrf2 axis therefore drives a hormetic response that is required for optimal cardiac mitochondrial and contractile function during physiological exercise.
DOI: https://doi.org/10.7554/eLife.41044.001

*For correspondence:
ajay.shah@kcl.ac.uk (AMS);
min.zhang@kcl.ac.uk (MZ)

## Introduction

Regular physical exercise has widespread beneficial effects on multiple body systems in healthy individuals and increases quality of life and lifespan. It is one of the most effective ways of preventing cardiovascular disease through reduction in risk factors such as obesity, hypertension and metabolic syndrome and by enhancing vascular function (*Mann and Rosenzweig, 2012*). Regular exercise also reduces morbidity and mortality in conditions such as chronic heart failure and dementia (*Sharma et al., 2015*). Fundamental to these beneficial effects of exercise is the capacity of the heart to substantially increase its performance and output during exercise, safely and without detriment to the heart itself. The intrinsic mechanisms that underlie the physiological adaptation of the heart to regular exercise are therefore crucial to its overall beneficial effects (*Vega et al., 2017*).

Previous studies show that intrinsic pathways that mediate cardiac adaptive responses to exercise are triggered rapidly, even after just 1–3 bouts of consecutive exercise, and are sufficiently robust to provide significant cardioprotection (*Demirel et al., 2001*; *Powers et al., 2008*). Among the candidate pathways involved in the adaptive response to acute exercise are those that maintain a balance

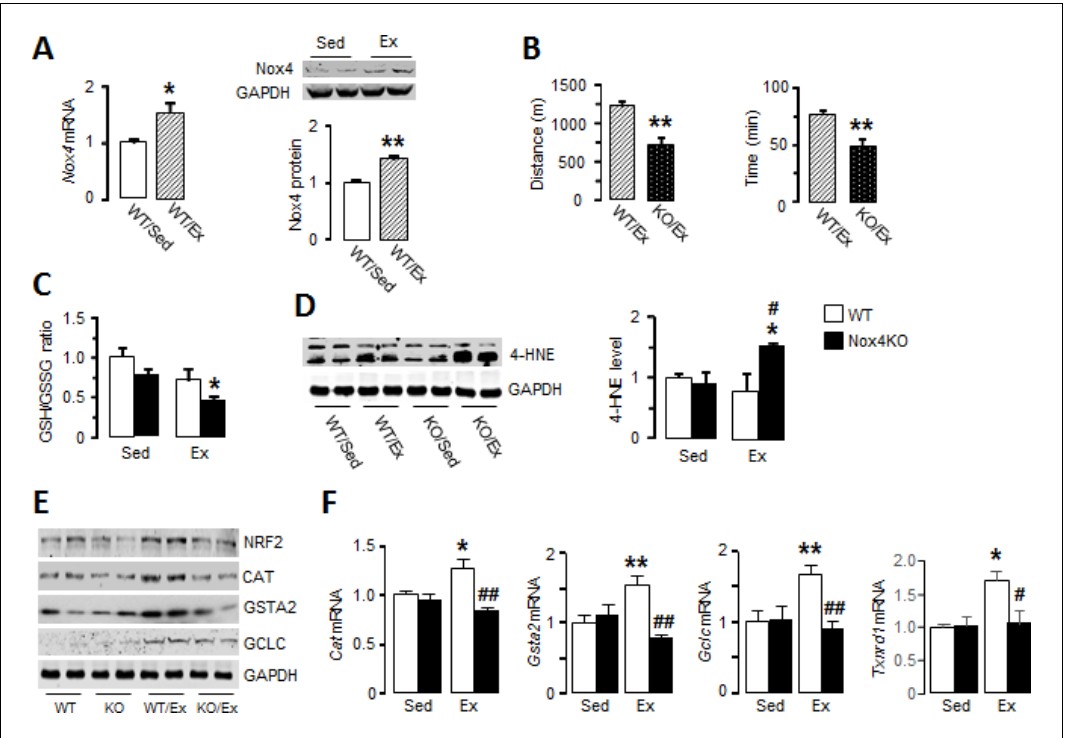

**Figure 1.** Nox4 mediates myocardial Nrf2 activation and is essential in the acute response to exercise. (**A**) Changes in myocardial Nox4 mRNA and protein levels after acute moderate exercise (Ex) compared to sedentary controls (Sed). *p<0.05, **p<0.01, 2-tailed t-test (n = 4–6/group). (**B**) Exercise capacity of *Nox4*-null mice (Nox4KO) and littermate wild-types (WT) measured by maximal running distance and running time. **p<0.01, 2-tailed t-test (n = 5–8/group). (**C**) Ratio of glutathione/glutathione disulfide (GSH/GSSG) in Nox4KO and WT mouse hearts before (Sed) and after exercise (Ex). n = 6–9/group. (**D**) Protein levels of 4-hydroxynonenal (4-HNE) adducts in the heart. n = 3–4/group. (**E and F**) Protein and mRNA levels of major Nrf2 targets. n = 3–6/group *p<0.05, **p<0.01 vs respective sedentary controls (Sed); #p<0.05, ##p<0.01 vs WT/Ex, 1-way ANOVA followed by Tukey post hoc analysis. CAT: catalase, GSTA2: glutathione S-transferase A2, GCLC: glutamate cysteine ligase catalytic subunit, TXNRD1: thioredoxin reductase 1.

DOI: https://doi.org/10.7554/eLife.41044.002

The following figure supplements are available for figure 1:

**Figure supplement 1.** Full images of Western blots in *Figure 1*.
DOI: https://doi.org/10.7554/eLife.41044.003
**Figure supplement 2.** Myocardial Nox2 is unchanged during acute exercise.
DOI: https://doi.org/10.7554/eLife.41044.004
**Figure supplement 3.** Quantification of Western blots in *Figure 1E*.
DOI: https://doi.org/10.7554/eLife.41044.005

between reactive oxygen species (ROS) and endogenous antioxidant defences. The maintenance of a physiological redox state is crucial for all cellular functions and is likely to be especially important when cardiac workload and the activity of metabolic pathways that support it are changing rapidly during exercise. Myocardial ROS levels rise transiently during acute exercise (*Muthusamy et al., 2012*), which is thought to be related to enhanced activity of the mitochondrial respiratory chain during increases in cardiac contractility. Excessive levels of ROS (oxidative stress) could cause damage to membranes, proteins and DNA and lead to detrimental consequences such as mitochondrial, metabolic and cellular dysfunction. However, these potentially detrimental effects are limited by endogenous antioxidant defences, and exercise has been found to increase antioxidant capacity in the heart and skeletal muscle (*Powers et al., 2014*; *Radak et al., 2017*). Interestingly, low-level increases in ROS may themselves induce a response that counteracts oxidative stress (termed hormesis), which represents a physiological adaptive mechanism (*Radak et al., 2017*). However, the sources of ROS that may be involved and their regulatory roles in this process remain elusive.

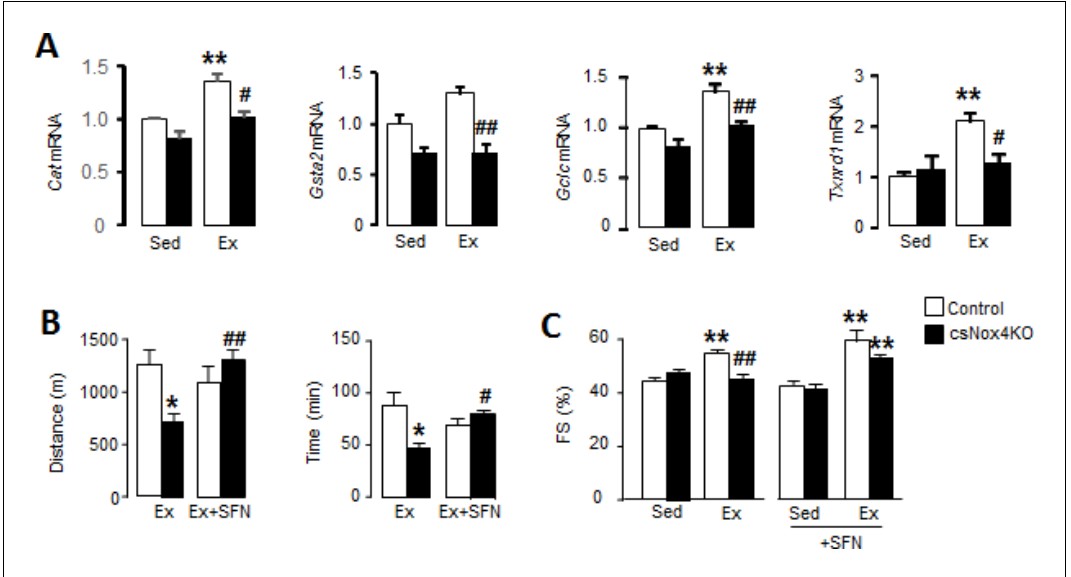

**Figure 2.** Cardiac Nox4 deficiency impairs heart function and exercise capacity due to lack of Nrf2 activation. (**A**) Expression of Nrf2-related genes in hearts of cardiomyocyte-specific Nox4KO (csNox4KO) and littermate controls after exercise (Ex). n = 3–4/group. **p<0.01 vs respective sedentary controls (Sed); #p<0.05, ##p<0.01 vs Control/Ex. (**B**) Maximal exercise distance and time. n = 3–5/group. *p<0.05 vs Control/Ex; #p<0.05, ##p<0.01 vs csNox4KO/Ex, 1-way ANOVA followed by Tukey post hoc analysis. (**C**) Fractional shortening (FS) evaluated by conscious echocardiography in control and csNox4KO mice immediately after the exercise capacity test. Some animals were treated with sulforaphane (SFN) prior to exercise. n = 4–12/group. **p<0.01 vs Sed; ##p<0.01 vs Control/Ex, 2-way ANOVA followed by Tukey post hoc analysis. *Cat*: catalase, *Gsta2*: glutathione S-transferase A2, *Gclc*: glutamate cysteine ligase catalytic subunit, *Txnrd1*: thioredoxin reductase 1.
DOI: https://doi.org/10.7554/eLife.41044.006

The following figure supplement is available for figure 2:

**Figure supplement 1.** csNox4KO mice exhibit reduced cardiac function at peak exercise.
DOI: https://doi.org/10.7554/eLife.41044.007

Nuclear factor E2-related factor 2 (Nrf2) is a transcription factor which is a master regulator of cellular redox balance. It binds to antioxidant response elements (AREs) in the promoter regions of its target genes which include enzymes involved in glutathione biosynthesis and maintenance such as glutamate-cysteine ligase catalytic subunit (GCLC) and glutathione reductase; antioxidant enzymes such as catalase (CAT), thioredoxin reductases (TRXRs) and haem oxygenase-1, and detoxification enzymes such as NAD(P)H quinone dehydrogenase 1 (NQO1) and glutathione S-transferases (GSTs) (*Niture et al., 2014*). Nrf2 normally undergoes rapid turnover through ubiquitination and proteosomal degradation. However, upon exposure to oxidants or electrophiles, its associated protein Keap1 is modified and enables Nrf2 to be stabilised and then accumulate in the nucleus to mediate gene transcription. Previous work implicates the upregulation of Nrf2 and endogenous antioxidant defences as an adaptive response to exercise in heart and skeletal muscle (*Done and Traustadóttir, 2016*; *Muthusamy et al., 2012*; *Oh et al., 2017*; *Wang et al., 2016*).

NADPH oxidase-4 (Nox4) is a member of the Nox family proteins, which generate ROS by catalysing electron transfer from NADPH to molecular $O_2$ (*Lassègue et al., 2012*). Unlike other ROS sources such as mitochondria, uncoupled nitric oxide synthases and xanthine oxidases, the Nox enzymes produce ROS as their primary function and have been shown to be involved in diverse redox signalling pathways. In contrast to other mammalian Nox isoforms, Nox4 is constitutively active and is regulated mainly by its abundance (*Zhang et al., 2013*). Previous studies show that cardiac Nox4 levels rise in response to diverse cellular stresses (*Lassègue et al., 2012*; *Zhang et al., 2010*), in part as a transcriptional response to ATF4 (*Santos et al., 2016*). In the disease setting of chronic pressure overload, the upregulation of Nox4 promotes adaptive cardiac remodelling through effects that include a preservation of myocardial capillary density (*Zhang et al., 2010*; *Zhang et al., 2018*) and

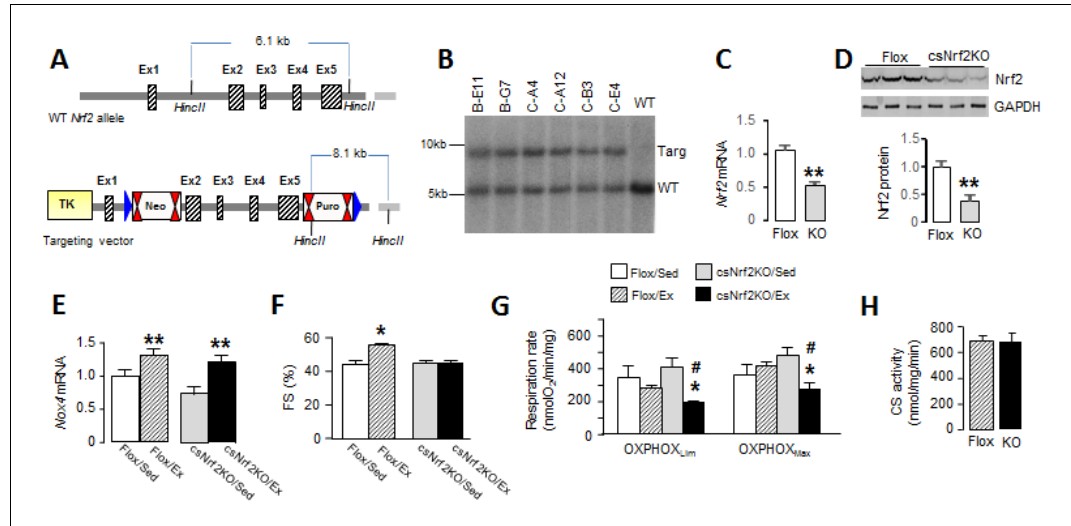

**Figure 3.** Cardiomyocyte-targeted *Nrf2* knockout mice have impaired mitochondrial function and reduced heart performance upon exercise. (**A**) Targeting strategy for generation of *Nrf2*<sup>fl/fl</sup> mice (Flox). The endogenous *Nrf2* locus is shown at the top and the targeting vector at the bottom. Exons 2–5 were flanked by LoxP sites which are represented by blue triangles; FRT (Neo) and F3 (Puro) sites are shown by double red triangles. Cre-mediated recombination deletes a 5.5 kb fragment including most of the open-reading frame and the 3′UTR. (**B**) Southern blot analysis of genomic DNA from selected ES cell clones, performed after excision at HincII sites, showing correct 3′ homologous recombination in all clones. WT indicates wild-type. (**C**) Nrf2 mRNA levels (n = 6/group) and (**D**) protein levels (n = 3/group) in the hearts of csNrf2KO and Flox control mice. **p<0.01, 2-tailed t-test. (**E**) Increase in *Nox4* mRNA levels in csNrf2KO and control mouse hearts after acute moderate exercise. **p<0.01 vs respective sedentary controls, 2-tailed t-test (n = 6/group). (**F**) Fractional shortening (FS) by echocardiography immediately after the exercise capacity test. *p<0.05 vs sedentary (Sed) Flox control mice, 1-way ANOVA followed by Tukey post hoc analysis (n = 6–7/group). (**G**) Respiration rates in isolated cardiac mitochondria of csNrf2KO and control mice after exercise. Both ADP-limited oxidative phosphorylation (OXPHOS<sub>Lim</sub>) and maximal oxidative capacity (OXPHOS<sub>Max</sub>) were measured (see Materials and methods for details). *p<0.05 vs csNrf2KO/Sed, # p<0.05 Flox/Ex, 1-way ANOVA followed by Tukey post hoc analysis (n = 4–6/group). (**H**) Citrate synthase (CS) activity in the heart. n = 4–6/group.

DOI: https://doi.org/10.7554/eLife.41044.008

The following figure supplements are available for figure 3:

**Figure supplement 1.** Identification of csNrf2KO mouse.

DOI: https://doi.org/10.7554/eLife.41044.009

**Figure supplement 2.** csNrf2KO mice exhibit normal cardiac morphology and function at baseline.

DOI: https://doi.org/10.7554/eLife.41044.010

**Figure supplement 3.** Respiratory rates in isolated cardiac mitochondria of csNrf2KO mice after exercise.

DOI: https://doi.org/10.7554/eLife.41044.011

an Nrf2-dependent enhancement of myocardial redox state (*Smyrnias et al., 2015*). However, it is not known whether Nox4 plays any physiological role in the heart.

Here, we report that cardiomyocyte Nox4 is an essential mediator of the physiological activation of the Nrf2 pathway during acute exercise, triggering an adaptive response that preserves redox balance, mitochondrial function and exercise performance. Our findings identify a novel physiological pathway that is required for the heart to safely and efficiently support physical exercise.

## Results

### Nox4 is required for a physiological exercise response

It was previously shown that Nox4 mRNA and protein levels are low in the adult mouse heart but are upregulated by chronic hemodynamic overload (*Zhang et al., 2010*). We first analysed whether myocardial Nox4 levels change in response to physiological exercise. In mice subjected to 2 bouts of

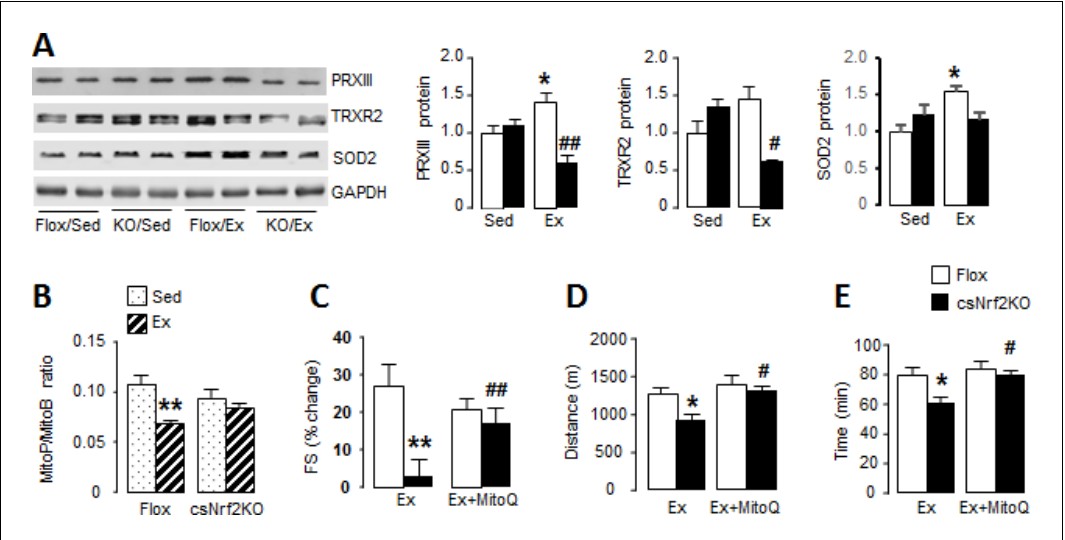

**Figure 4.** Cardiomyocyte Nrf2 is required for increase in mitochondrial antioxidant capacity with exercise. (**A**) Western blots for peroxiredoxin III (PRXIII), thioredoxin reductase-2 (TRXR2) and superoxide dismutase-2 (SOD2) in heart of csNrf2KO and Flox controls after acute exercise (Ex). Mean data from n = 4–6/group are shown on the right. *p<0.05 vs respective sedentary controls (Sed); #p<0.05, ##p<0.01 vs Flox/Ex, 1-way ANOVA followed by Tukey post hoc analysis. (**B**) In vivo mitochondrial $H_2O_2$ levels assessed by the MitoP/MitoB ratio in hearts of csNrf2KO mice and controls after acute exercise. **p<0.01 vs Flox/Sed, 1-way ANOVA followed by Tukey post hoc analysis (n = 5–6/group). (**C**) MitoQ treatment improved cardiac contractile performance at peak exercise in csNrf2KO mice. n = 6–7/group. (**D**) and (**E**) The effect of MitoQ treatment on maximal running distance and time. n = 7–11/group. *p<0.05, **p<0.01 vs Flox/Ex, #p<0.05, ##p<0.01 vs csNrf2KO/Ex, 2-way ANOVA followed by Tukey post hoc analysis.

DOI: https://doi.org/10.7554/eLife.41044.012

The following figure supplements are available for figure 4:

**Figure supplement 1.** Full images of Western blots in *Figure 4*.
DOI: https://doi.org/10.7554/eLife.41044.013
**Figure supplement 2.** Nox4 is required for increase in mitochondrial antioxidant capacity with exercise.
DOI: https://doi.org/10.7554/eLife.41044.014
**Figure supplement 3.** Mitochondrial-targeted antioxidant MitoQ improves cardiac response to exercise in csNrf2KO.
DOI: https://doi.org/10.7554/eLife.41044.015

1500 m moderate intensity treadmill exercise on consecutive days, there was a significant increase in myocardial Nox4 mRNA and protein levels as compared to sedentary mice (*Figure 1A* and *Figure 1—figure supplement 1A*). No change was observed in the expression levels of Nox2, the other main Nox isoform that is expressed in the heart (*Figure 1—figure supplement 2A*) or in Nox2 activation as assessed by the membrane translocation of its essential regulatory subunit, p47[phox], after physiological exercise (*Figure 1—figure supplement 2B*). We next conducted a maximal exercise capacity test in which *Nox4*-null mice and matched wild-type littermates were run to exhaustion on day 3. This test revealed that *Nox4*-null mice had a maximal running distance that was ~60% of that in wild-type controls and a maximal running time that was ~65% of that in controls (*Figure 1B*). These results indicate that Nox4 has an essential role in facilitating acute exercise in healthy mice.

## Nox4 mediates exercise-induced activation of Nrf2 in the heart

To investigate potential mechanisms underlying the lower exercise capacity in *Nox4*-null mice, we first assessed exercise-induced changes in myocardial redox state. The ratio of glutathione/glutathione disulfide (GSH/GSSG), a sensitive marker of cellular redox state, was significantly lower after treadmill running in *Nox4*-null compared to sedentary mouse hearts (*Figure 1C*). Consistent with a more oxidized cellular environment, the levels of 4-hydroxynonenal (4-HNE) adducts as a marker of lipid peroxidation due to oxidative stress were significantly increased in the myocardium of *Nox4*-

null mice after exercise (*Figure 1D* and *Figure 1—figure supplement 1B*). These results suggest that the loss of exercise-induced increase in Nox4 levels results in elevated oxidative stress in the heart after acute exercise.

Since Nrf2 is an important regulator of exercise-induced redox adaptations (*Done and Trausta-dóttir, 2016*), we assessed whether the absence of Nox4 affects Nrf2 levels during acute exercise. Myocardial Nrf2 protein levels were significantly elevated after acute exercise in wild-type mice but this response was absent in *Nox4*-null mice (*Figure 1E* and *Figure 1—figure supplement 1C* and *Figure 1—figure supplement 3*). In parallel with the rise in Nrf2, wild-type mouse hearts showed increases in transcript levels of the major Nrf2 target genes *Cat*, *Gsta2*, *Gclc* and *Txnrd1* (*Figure 1F*), as well as increases at protein level (*Figure 1E* and *Figure 1—figure supplement 1C* and *Figure 1—figure supplement 3*). However, there was no increase in levels of these antioxidant defence proteins after exercise in *Nox4*-null mice (*Figure 1E* and *Figure 1—figure supplement 1C* and *Figure 1—figure supplement 3*). No differences were observed between wild-type and *Nox4*-null mouse hearts at baseline. Therefore, physiological exercise-induced activation of the Nrf2 pathway in the heart appears to be dependent on endogenous Nox4.

## Cardiac Nox4 deficiency impairs heart function and exercise capacity due to lack of Nrf2 activation

To exclude potential systemic effects of global Nox4 deficiency and to specifically establish the functional role of cardiomyocyte Nox4 during exercise, we next studied inducible cardiomyocyte-specific Nox4 knockout (csNox4KO) mice (*Zhang et al., 2018*). csNox4KO mice were compared to Cre-negative Flox control mice. Similar to *Nox4*-null mice, the csNox4KO animals failed to display an increase in Nrf2 target genes upon exercise (*Figure 2A*), consistent with a lack of Nrf2 activation. There was no difference between csNox4KO mice and matched control mice in the absence of exercise. The csNox4KO mice achieved a significantly shorter running distance and exercise time than matched controls during an exercise capacity test (*Figure 2B*), with the magnitude of defect being very similar to that observed in global *Nox4*-null mice. We assessed cardiac contractile function by echocardiography in conscious mice, immediately upon cessation of the exercise capacity test. While the left ventricular fractional shortening (FS) increased by 24% in the control group at peak exercise, there was a 6% decrease in FS in the csNox4KO group (*Figure 2C*). The increase in FS in control animals was driven by a decrease in end-systolic dimension, indicative of increased contractility, which was absent in csNox4KO mice whereas there was no difference in the heart rate response between groups (*Figure 2—figure supplement 1A*).

To assess whether the defect in cardiac performance and exercise capacity in csNox4KO mice is related to the absence of Nrf2 activation, we treated both csNox4KO and control animals with an Nrf2 activator, sulforaphane, administered for 2 days prior to the exercise capacity test. The treatment with sulforaphane fully restored the exercise-induced increase in cardiac performance and maximal exercise capacity in csNox4KO mice (*Figure 2*, B and C, and *Figure 2—figure supplement 1, B* and C). Taken together, these results suggest that cardiomyocyte Nox4-dependent activation of Nrf2 is essential for the normal physiological increase in cardiac performance during acute exercise, which is required to achieve maximal exercise capacity.

## Cardiomyocyte Nrf2 is required for optimal increments in heart performance during exercise

To more directly define the role of cardiac Nrf2 in the physiological response to exercise, we generated a new mouse line with a cardiomyocyte-specific deletion of Nrf2 (csNrf2KO mice, *Figure 3, A and B*). The csNrf2KO mice showed a significant cardiac-specific decrease in *Nrf2* mRNA levels in the heart with no change in other organs (*Figure 3, C* and *Figure 3—figure supplement 1B*). There was an approximately 60% reduction in Nrf2 protein in the heart (*Figure 3D* and *Figure 3—figure supplement 1A*), with the residual levels in csNrf2KO mice probably attributable to expression in non-myocytes. There were no differences in baseline cardiac size or function between csNrf2KO mice and control *Nrf2*fl/fl littermates (*Figure 3—figure supplement 2*). Myocardial levels of Nox4 were similar in csNrf2KO mice and control littermates at baseline and increased to a similar extent in both genotypes after acute exercise (*Figure 3E*), confirming that Nox4 lies upstream of Nrf2. However, when the two groups were subjected to a maximal exercise capacity test, the csNrf2KO mice

exhibited a significant impairment of cardiac function (*Figure 3F*) and a reduced running distance and time as compared to control littermates (*Figure 4, D and E*). These results indicate a critical role of cardiomyocyte Nrf2 in facilitating an optimal cardiac functional response to acute exercise.

## Mitochondrial function is impaired in csNrf2KO mice after exercise

We sought to identify mechanisms that account for the beneficial effects of Nrf2 activation on the acute response to exercise. Since optimal mitochondrial function is essential to support an increase in cardiac output during exercise and depends upon a preserved redox state, we investigated mitochondrial respiratory function in csNrf2KO mice and control littermates. Cardiac mitochondria were isolated from csNrf2KO mice and control littermates immediately upon completion of the exercise capacity test and studied by high-resolution respirometry. There were no significant differences between sedentary csNrf2KO mice and littermate controls in respiratory capacity measured as ADP-limited and maximal oxidative phosphorylation ($OXPHOS_{Lim}$ and $OXPHOS_{Max}$, respectively) (*Figure 3G*). However, after exercise, both sub-maximal and maximal respiratory capacity were significantly lower in csNrf2KO mice compared to littermate sedentary controls (*Figure 3G*). Neither mitochondrial efficiency (P:O ratio) nor respiratory coupling ratio (an index of the tightness of coupling between respiration and phosphorylation) were altered by genotype or exercise (*Figure 3—figure supplement 3A and B*). Mitochondrial membrane integrity as assessed by the response to cytochrome C in the oxygraph protocol was similar among groups (*Figure 3—figure supplement 3C*). Mitochondrial mass, as estimated by the total citrate synthase activity, was also similar between genotypes (*Figure 3H*). Results from the substrate-uncoupler-inhibitor-titration (SUIT) protocol confirmed impaired mitochondrial oxidative phosphorylation capacity in exercised csNrf2KO hearts when using fatty acids, complex I and complex I I substrates (*Figure 3—figure supplement 3D*)

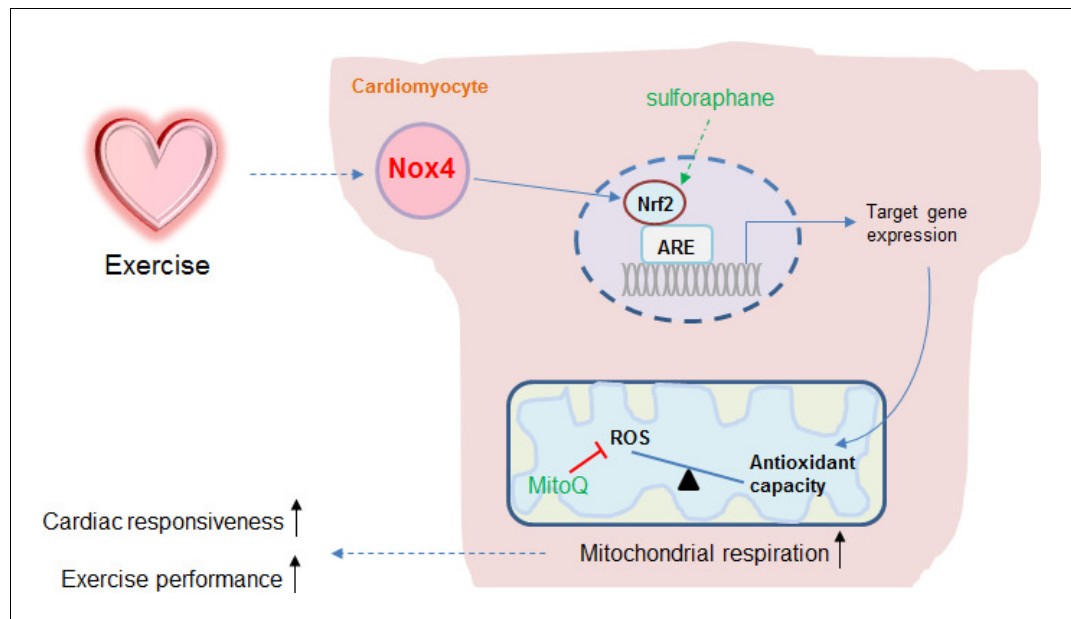

**Figure 5.** Schematic illustrating the role of the cardiomyocyte Nox4-Nrf2 pathway in enhancing mitochondrial and cardiac contractile function during acute physiological exercise. Cardiac Nox4 levels increase in response to acute exercise and induce an activation of Nrf2. The Nrf2-dependent gene program includes endogenous antioxidants that are critical in maintaining mitochondrial redox balance during peak exercise. The maintenance of mitochondrial redox balance is essential to achieve maximal mitochondrial respiration and cardiac contractile function at peak exercise. The sites of action of the Nrf2 activator sulforaphane and the mitochondria-targeted antioxidant MitoQ are shown.
DOI: https://doi.org/10.7554/eLife.41044.016

## Nox4 and Nrf2 are required for exercise-induced increases in mitochondrial antioxidants

In view of the impaired mitochondrial function at peak exercise in csNrf2KO mice, we hypothesised that there may be a defect in the redox homeostasis required for normal respiratory function (*Holmström et al., 2016*). We found that acute exercise resulted in significant increases in the levels of the major mitochondrial antioxidants peroxiredoxin III (PRXIII), thioredoxin reductase-2 (TRXR2) and superoxide dismutase 2 (SOD2) in littermate control mouse hearts (*Figure 4A* and *Figure 4—figure supplement 1*). These increases were, however, abolished in the hearts of exercised csNrf2KO mice although there were no differences between sedentary csNrf2KO and control mice. A similar abolition of exercise-induced increase in these mitochondrial antioxidants was also found in Nox4KO hearts (*Figure 4—figure supplement 2*). These findings indicate that the Nox4/Nrf2 axis is required for exercise-induced increases in mitochondrial antioxidants.

To investigate the impact of the altered mitochondrial antioxidants in csNrf2KO mouse hearts with exercise, we assessed mitochondrial hydrogen peroxide ($H_2O_2$) levels in vivo using injection of the mitochondrial-targeted probe MitoB (*Cochemé et al., 2012*). This probe is rapidly concentrated in mitochondria where it reacts with $H_2O_2$ to form MitoP; quantification of the MitoP/MitoB ratio by LC-MS provides a sensitive measure of mitochondrial $H_2O_2$ levels (*Logan et al., 2014*). We found in normal mouse hearts that mitochondrial $H_2O_2$ levels were significantly lower immediately upon cessation of peak exercise than in the non-exercising state (*Figure 4B*) – consistent with a reduction in ROS at peak exercise due to an enhanced antioxidant state. In marked contrast, there was no decrease in mitochondrial $H_2O_2$ levels with exercise in csNrf2KO mice, in line with the failure to enhance mitochondrial antioxidants. There were no significant differences between genotypes in sedentary mice.

## Supplementation with a mitochondria-targeted antioxidant rescues the physiological cardiac response and exercise capacity in csNrf2KO mice

Finally, to confirm that the defect in mitochondrial redox balance is responsible for the impaired cardiac performance and exercise capacity in csNrf2KO mice, we performed rescue experiments in which mice were treated with a small molecule mitochondria-targeted antioxidant, MitoQ, for 2 days prior to the exercise capacity test (*Smith and Murphy, 2010*). We found that MitoQ supplementation markedly improved cardiac performance at peak exercise in csNrf2KO mice and fully restored the maximal exercise capacity in these animals (*Figure 4, C to E*, and *Figure 4—figure supplement 3*). MitoQ did not have a significant additional effect on exercise performance in control mice.

## Discussion

The multiple beneficial effects of regular exercise depend upon the ability of the healthy heart to safely and consistently increase its contractile performance during the periods when cardiac output and overall body oxygen consumption are elevated. It has long been recognized that adaptations that facilitate the heart's ability to safely undertake repetitive exercise begin to be triggered very rapidly, after just 1–3 episodes of consecutive exercise. One of the most important of such adaptations may be the maintenance of a physiological redox balance, given that this is crucial for mitochondrial energy metabolism and heart contractile performance (*Powers et al., 2014*). In this study, we identify cardiomyocyte Nox4 as the physiological driver of exercise-induced upregulation of endogenous antioxidants, serving to preserve mitochondrial redox state and respiratory function and thereby facilitate optimal cardiac contractile performance and exercise capacity (*Figure 5*). We show that consecutive episodes of exercise upregulate myocardial Nox4 levels and lead to the activation of an Nrf2-dependent transcriptional program that boosts endogenous antioxidants and mitigates oxidative stress during exercise. The cardiomyocyte-specific loss of either Nox4 or Nrf2 results in a significant reduction in exercise cardiac performance and maximal exercise capacity in healthy adult mice. The exercise defect in cardiomyocyte-specific *Nox4*-deficient mice is corrected by a small molecule Nrf2 activator, sulforaphane, placing Nrf2 activation downstream of Nox4. On the other hand, the defect in exercise capacity in cardiomyocyte-specific *Nrf2*-deficient mice can be rescued by a mitochondria-targeted antioxidant, MitoQ, confirming that it relates to a disruption of

mitochondrial redox balance. Taken together, these data identify a crucial Nox4-dependent physiological pathway that mediates cardiac adaptation to repeated acute exercise.

The activity of Nox4 depends on its protein abundance but its levels in the healthy adult heart are very low (*Brewer et al., 2011*). Previous studies show that an increase in Nox4 abundance is induced by diverse pathological stresses such as hypoxia, ischaemia and haemodynamic overload (*Zhang et al., 2010*). In the current study, cardiac Nox4 mRNA and protein levels were significantly increased after two bouts of moderate intensity exercise on consecutive days. The precise stimulus for the increase in Nox4 levels with exercise was not studied. However, the transcriptional induction of Nox4 by ATF4 was recently identified as an important mechanism for its stress-induced upregulation (*Santos et al., 2016*), and it is of interest that exercise is reported to upregulate ATF4 in human skeletal muscle (*Drummond et al., 2011*). An increase in levels of Nox4 as a ROS-producing protein complex might be envisaged to generate detrimental oxidative stress. However, it has been found that Nox4 mediates protective effects in pathological stress situations, such as chronic hemodynamic overload or nutrient starvation in the heart (*Zhang et al., 2010*; *Sciarretta et al., 2013*) and angiotensin II-induced remodelling or atherosclerosis in the vasculature (*Craige et al., 2015*; *Schröder et al., 2012*; *Schürmann et al., 2015*). These effects typically involve localized redox signalling without overt oxidative stress. In fact, Nox4 was found to specifically activate Nrf2 during load-induced cardiac remodeling (*Smyrnias et al., 2015*) and during angiotensin II-induced vascular remodeling (*Schröder et al., 2012*). Here, we show that myocardial Nox4 mediates a similar activation of Nrf2 during acute exercise – representing to the best of our knowledge the first physiological function of cardiac Nox4 that has so far been identified. The Nox4-dependent activation of Nrf2 was required for optimal cardiac function and running capacity during maximal exercise, as evidenced by the restoration of normal exercise responses in *Nox4*-deficient animals that were treated with the Nrf2 activator, sulforaphane. Notably, we observed no increase in cardiac contractility as assessed by fractional shortening in *Nox4*-deficient mice at peak exercise whereas there was a 24% increase in wild-type animals.

Nrf2 is a master transcriptional regulator of antioxidant and stress-defense proteins and previous studies found that it is activated in the heart upon acute exercise (*Muthusamy et al., 2012*). A canonical mechanism for Nrf2 activation is in response to redox signals which result in its post-translational stabilisation. In principle, Nrf2 activation could be activated by diverse sources of ROS, such as mitochondria or different Nox isoforms. However, here we find that endogenous Nox4 is an essential and indispensable regulator of myocardial Nrf2 activation during physiological acute exercise. As such, the acute increase in myocardial Nrf2 protein levels and its major downstream targets (such as antioxidants and GSH-generating enzymes) was markedly inhibited in the hearts of *Nox4*-null mice or csNox4KO mice. Furthermore, this was accompanied by a significant redox imbalance as assessed by the GSH/GSSG ratio and by evidence of oxidative stress as assessed by the levels of 4-HNE adducts, along with a reduced maximal exercise capacity. Therefore, the absence of Nox4 resulted in redox imbalance during exercise as a result of impaired Nrf2 activation and impaired antioxidant reserve, and led to a significant impairment of exercise performance. The increased ROS levels in this setting most likely emanated from mitochondria (*Radak et al., 2013*; *Saborido et al., 2011*), as evidenced by experiments using the mitochondria-targeted redox probe MitoB and the mitochondrial-targeted antioxidant MitoQ.

To more specifically establish the role of cardiomyocyte Nrf2 in exercise-induced cardiac adaptation and determine how it enhances exercise performance, we developed a new mouse model with cardiomyocyte-specific deficiency of Nrf2. These animals had a normal heart structure and function at baseline but displayed a significant impairment of cardiac performance and running capacity upon maximal exercise, similar to the phenotype observed in *Nox4*-deficient animals. The csNrf2KO mice, however, developed a similar elevation in cardiac Nox4 levels upon exercise to that observed in control mice, consistent with Nox4 being upstream of Nrf2. Investigation of cardiac mitochondrial function in csNrf2KO mice revealed that the respiratory capacity at maximal exercise was significantly impaired compared to control animals, without any differences in mitochondrial mass or membrane integrity. Although emerging data suggest that Nrf2 can have multi-faceted effects on mitochondrial metabolism (*Dinkova-Kostova and Abramov, 2015*; *Holmström et al., 2016*; *Merry and Ristow, 2016b*), we found no difference in cardiac mitochondrial function between sedentary csNrf2KO and control mice, indicating that the difference between genotypes was specific to exercise. Further analyses revealed that the csNrf2KO mouse hearts failed to increase the mitochondrial antioxidants

PRXIII, TRXR2 and SOD2 upon exercise, in contrast to control animals. In parallel, we found that mitochondrial ROS levels were lower after exercise in control mice (consistent with increased antioxidant levels) but that this response was absent in csNrf2KO mouse hearts. Taken together, these results indicate that an Nrf2-dependent induction of mitochondrial antioxidants is critical to maintain redox balance during acute exercise. Our study also shows that cardiomyocyte Nox4 is the essential activator of this adaptive response, resulting in the maintenance of optimal mitochondrial respiration and cardiac function during maximal exercise (*Figure 5*).

Whether antioxidant supplementation improves exercise performance is a controversial question (*Merry and Ristow, 2016a*; *Ristow et al., 2009*). A reduction in excessive oxidative stress resulting from mitochondrial ROS production is likely to be beneficial but if ROS-mediated adaptive responses are inhibited by antioxidants, this might be detrimental (*Lauer et al., 2005*; *Sachdev and Davies, 2008*). Previous studies have identified several ROS-mediated adaptive responses during exercise. The current study suggests that what is ultimately most important is the redox balance during exercise, especially at the level of the mitochondria. Targeted antioxidants such as MitoQ that primarily reduce mitochondrial ROS levels (*Oyewole and Birch-Machin, 2015*) may be more likely to have beneficial effects than non-specific antioxidants that might inhibit Nrf2 activation. Interestingly, we did not find any beneficial effects of either sulforaphane or MitoQ in the control animal groups, suggesting that there is limited scope for further enhancement of the adaptive exercise redox response in healthy adult mice with normal Nox4/Nrf2 activation, at least in the relatively acute setting. This is consistent with prior findings that MitoQ did not enhance aerobic training adaptations or skeletal muscle oxidative capacity in young healthy men (*Shill et al., 2016*). However, it is possible that the Nox4/Nrf2/mitochondrial antioxidant pathway could be usefully augmented in older individuals or those with disease-related impairment of this response.

Taken together, these data identify cardiomyocyte Nox4 as a crucial physiological mediator of Nrf2 activation, triggering an adaptive response that maintains redox state, mitochondrial function and cardiac contractile performance to support normal physical exercise.

# Materials and methods

## Key resources table

| Reagent type (species) or resource | Designation | Source or reference | Identifiers | Additional information |
|---|---|---|---|---|
| Genetic reagent (*M. musculus*) | *Nox4*-null | PMID: 20921387 | | Dr. Schröder (Institut für Kardiovaskuläre Physiologien, Goethe-Universität) |
| Genetic reagent (*M. musculus*) | cardiomyocyte-specific Nox4 knockout | PMID: 29040462 | | Dr. Shah (King's College London British Heart Foundation Centre of Excellence) |
| Genetic reagent (*M. musculus*) | *Nrf2*fl/fl | in this paper | | Dr. Shah (King's College London British Heart Foundation Centre of Excellence) |
| Genetic reagent (*M. musculus*) | Myl2tm1(cre)Krc | The Jackson Laboratory (Stock No: 029465) | RRID:IMSR_JAX:029465 | |
| Antibody | Anti-GSTA2 (rabbit polyclonal) | Sigma-Aldrich Cat# SAB1401163 | RRID:AB_10609834 | |
| Antibody | Mouse Anti-GAPDH Monoclonal Antibody | Sigma-Aldrich Cat# G8795 | RRID:AB_1078991 | |
| Antibody | Anti-4 Hydroxynonenal Rabbit Polyclonal Antibody | Abcam Cat# ab46545 | RRID:AB_722490 | |
| Antibody | Anti-MnSOD Rabbit Polyclonal Antibody | Abcam Cat# ab13533 | RRID:AB_300434 | |
| Antibody | Anti-GCLC Rabbit Polyclonal Antibody | Novus Cat# NBP1-49762 | RRID:AB_10011848 | |

*Continued on next page*

*Continued*

| Reagent type (species) or resource | Designation | Source or reference | Identifiers | Additional information |
|---|---|---|---|---|
| Antibody | Anti-Catalase Rabbit Polyclonal Antibody | Millipore Cat#219010 | RRID:AB_211660 | |
| Antibody | TrxR2 Goat Polyclonal Antibody | Santa Cruz Biotechnology Cat# sc-46278 | RRID:AB_2210408 | |
| Antibody | PRX III (C-14) Goat Polyclonal Antibody | Santa Cruz Biotechnology Cat# sc-23973 | RRID:AB_2237431 | |
| Antibody | NFE2L2-Rabbit Polyclonal Antibody | Santa Cruz Biotechnology Cat# sc-13032 | RRID:AB_2263168 | |
| Antibody | NFE2L2-Rabbit Polyclonal Antibody | Santa Cruz Biotechnology Cat# sc-722, | RRID:AB_2108502 | |
| Antibody | VE-cadherin (F-8) monoclonal antibody | Santa Cruz Biotechnology Cat# sc-9989 | RRID:AB_2077957 | |
| Antibody | Anti-p47-phox (mouse) antibody | Millipore Cat# 07–500 | RRID:AB_310668 | |
| Antibody | gp91phox mouse monoclonal antibody | BD Biosciences Cat# 611414 | RRID:AB_398936 | |
| Antibody | Nox4 rabbit polyclonal antibody | PMID: 18467643 | | *Anilkumar et al. (2008)* |

## Animals

The studies complied with the UK Home Office Guidance on the Operation of the Animals (Scientific Procedures) Act, 1986 and King's College London institutional guidelines. All the mouse lines studied were on a C57BL/6 background. Global Nox4KO mice were described previously (*Zhang et al., 2010*) and were compared to littermate wild-type controls. Inducible cardiomyocyte-specific Nox4KO mice (csNox4KO) were generated by crossing $Nox4^{fl/fl}$ female mice with male α-MHC-Mer-CreMer mice (*Zhang et al., 2018*). Tamoxifen was administered by i.p. injection (20 mg/kg/day) for 5 days to induce Cre expression. csNox4KO mice were compared with tamoxifen-treated Cre-negative Flox littermates, with the groups subjected to the exercise regime 1 week after completion of tamoxifen treatment. Tamoxifen treatment itself had no effect on cardiac function (data not shown).

The generation of $Nrf2^{fl/fl}$ mice was commissioned from Taconic-Artemis (Germany). In brief, the targeting vector was generated using BAC clones from the C57BL/6 RPCI-23 BAC library. The LoxP sites flanked exons 2 and 5 and the vector included an FRT (neomycin resistance) and F3 (puromycin resistance) site. The targeting vector was transfected into C57BL/6 Tac embryonic stem (ES) cell lines and recombinant clones were identified by PCR and Southern blotting, then injected into blastocysts. Heterozygous Floxed mice obtained from germline chimeras were bred with C57BL/6 Flp-deletor mice to excise the resistance cassettes. Mice were backcrossed onto a C57BL/6 background. Cardiomyocyte-specific Nrf2 knock-out mice (designated as csNrf2KO) were obtained by crossing $Nrf2^{fl/fl}$ mice with Cre-recombinase expressing mice under the control of a cardiomyocyte-specific Mlc2v promoter (Mlc2vCre). csNrf2KO and littermate $Nrf2^{fl/fl}$ controls were obtained by intercrossing $Mlc2vCre^{+/-}/Nrf2^{fl/fl}$ mice with $Nrf2^{fl/fl}$ mice.

## Acute exercise regimes

We studied female Nox4-null mice, csNox4KO mice or csNrf2KO mice and their respective littermate controls. Mice of age approximately 8 weeks were acclimatized to and trained on a 12.5° uphill Exer 3/6 open treadmill (Columbus Instruments) for 2 days. On both training days, mice ran for a total of 10 min, with a continual increase in speed up to a final speed of 15 m/min. Mice were then randomly assigned into a sedentary or exercise group. The exercise regime used in the study is a moderate intensity protocol as reported previously (*Hafstad et al., 2011*; *Muthusamy et al., 2012*). Briefly, on day 3 and 4 (experimental days 1 and 2), mice in the exercise group were subjected to a single bout of running at 15 m/min at a gradient of 12.5° for a total of 1500 m. Tissues for analyses of gene expression or immunoblotting were obtained after the second prolonged period of exercise.

Exercise capacity tests were performed on day 5 (experimental day 3). Mice started running at an initial treadmill speed of 15 m/min for 30 min. The speed was then increased at 1 m/min every 10 min for another 30 min, followed by a 1 m/min increase every 5 min until the mice were exhausted. Exhaustion was defined as the point at which mice spent more than 5 s on the electric shocker without attempting to resume running. The total running distance and running time were calculated for each mouse (*He et al., 2012*). In some experiments, sulforaphane (0.5 mg/kg i.p.) or MitoQ (4 mg/kg i.p.) were administered 24 hr and 2 hr prior to each exercise regime, respectively.

## Echocardiography

Immediately upon discontinuation of the exercise capacity test, cardiac function was evaluated in conscious mice by transthoracic echocardiography using a Vevo2100 system (Visualsonics, Canada) (*Zhang et al., 2015*). Echocardiography was undertaken on the same mouse prior to exercise as a baseline. Echocardiography was performed by an experienced operator blinded to the assignments. Images were acquired using M-mode at a depth setting of 11 mm and analyzed using Vevo software v1.7. Mice were acclimatized to conscious echocardiography over 2–3 days. Basal cardiac structure and function in sedentary csNrf2KO mice was assessed by echocardiography under 2% isoflurane anesthesia at heart rates > 400 bpm (*Zhang et al., 2015*).

## Real-time PCR

mRNA expression levels were quantified by real-time RT-PCR using SYBR Green on an Applied Biosystem PRISM 7700 machine. Delta delta Ct values were calculated with hypoxanthine-guanine phosphoribosyltransferase (HPRT) as an internal control. Primer sequences were (forward, reverse):

*Nfe2l2*, Nuclear factor, erythroid 2 like 2 (or Nrf2): CTACTCCCAGGTTGCCCACA, CGACTCATGGTCATCTACAAATGG;

*Nox4*, NADPH oxidase-4: CCGGACAGTCCTGGCTTATC, TGCTTTTATCCAACAATCTTCTTTT;

*Cat*, Catalase: CCCTTCGCAGCCATGTG, GCTGAGAAGCCTAAGAACGCAAT;

*Gsta2*, Glutathione S-transferase, alpha 2 (Yc2): GCTTGATGCCAGCCTTCTG, GGCTGCTGATTCTGCTCTTGA;

*Gclc*, Glutamate-cysteine ligase, catalytic subunit: GTTATGGCTTTGAGTGCTGCAT, ATCACTCCCCAGCGACAATC;

*Txnrd1*, thioredoxin reductase 1: GATGCACCAGGCAGCTTTG; TCTTCGACTTTCCAGCCATAGT;

*Hprt*, Hypoxanthine guanine phosphoribosyl transferase: CACAGGACTAGAACACCTGC, GCTGGTGAAAAGGACCTCT;

## Western blotting

Snap-frozen heart tissues were homogenized for immunoblotting. Antibodies used were: Nrf2, thioredoxin reductase-2, peroxiredoxin III and cadherin (Santa Cruz); Nox4 (*Anilkumar et al., 2008*); Nox2 (BD Biosciences); p47$^{phox}$, catalase (Millipore); glutathione S-transferase A2, 4-hydroxynonenal, superoxide dismutase-2 (Abcam); glutamate-cysteine ligase catalytic subunit (Novus biologicals). GAPDH (Sigma) was used as a loading control. Protein bands were visualized using enhanced chemiluminescence or fluorescence (Odyssey, LI-COR), and were quantified by densitometry.

Membrane proteins were obtained from heart homogenates using a standard centrifugation protocol (*Lu et al., 2009*). Cadherin and GAPDH were used as protein markers for the membrane and cytosolic fractions, respectively.

## Glutathione assay

The ratio of glutathione/glutathione disulfide (GSH/GSSG) in left ventricular homogenates was evaluated using a GSH-Glo assay kit (Promega) (*Brewer et al., 2011*), as a global measure of antioxidant status in the heart.

## Estimation of mitochondrial $H_2O_2$ in vivo

The mitochondrial-targeted probe, MitoB, was used to assess mitochondrial $H_2O_2$ levels in vivo, as previously described (*Cochemé et al., 2012*; *Logan et al., 2014*). MitoB can also react with peroxynitrite, but this reactive nitrogen species was not thought to contribute to the changes seen here. Briefly, 25 nmol MitoB in 50 μL saline was administered by tail vein injection 2 hr before the exercise

capacity test on experimental day 3. Hearts were excised immediately upon completion of exercise, snap frozen in liquid nitrogen and stored at −80°C. For analysis, around 50 mg tissue was homogenized and spiked with deuterated internal standards. MitoB and MitoP (produced by the reaction with $H_2O_2$) were extracted and quantified by liquid chromatography-tandem mass spectrometry relative to a standard curve (*Cochemé et al., 2012*; *Logan et al., 2014*).

## Mitochondrial respiratory function

Cardiac mitochondria were isolated immediately upon completion of the exercise capacity test and respiratory function was measured in an oxygraph (Oxygraph 2 k; Oroboros Instruments, Austria) as described previously (*Hafstad et al., 2013*; *Nabeebaccus et al., 2017*). Briefly, proton leak (Leak) respiration was measured with pyruvate (5 mM) and malate (1 mM) as substrates. ADP-limited respiration (oxidative phosphorylation, $OXPHOS_{Lim}$) was measured in the presence of ADP, 50 µM. Maximal mitochondrial respiratory capacity ($OXPHOS_{Max}$) was measured following the addition of ADP, 500 mM. Cytochrome C (10 µM) was added to assess mitochondrial membrane integrity ($OXPHOS_{Cyt}$). All respiration rates were normalized by the protein content. The phosphate/oxygen ratio (P/O ratio) was calculated from the ratio of molecules phosphorylated to each oxygen molecule consumed. Respiratory coupling ratio (RCR) was calculated as $OXPHOS_{Max}$/Leak.

In order to further study the mitochondrial function, a substrate-uncoupler-inhibitor-titration (SUIT) protocol was performed. Proton leak ($Leak_{FA}$) was measured in the presence of malate (0.5 mM), carnitine (5 mM) and palmityol-CoA (24 µM). Maximal respiratory capacity (oxidative phosphorylation, OXPHOS) was measured following the addition of a saturating amount of ADP (500 mM, $OXPHOS_{FA}$), saturating amounts of the complex I substrates pyruvate (10 mM) and glutamate (10 mM, $OXPHOS_{FA+CI}$) and the complex I I substrate succinate (10 mM, $OXPHOS_{FA+CI+CII}$). Maximal electron transport capacity (ETC) was measured following the addition of 0.75 µM carbonyl cyanide *m*-chlorophenyl hydrazone (CCCP) ($ETC_{FA+CI+CII}$). Rotenone (0.5 µM) was used to inhibit complex I ($ETC_{CII}$) and complex I I and III were inhibited by malonic acid (5 mM) and antimycin (2.5 µM), respectively (Residual oxygen consumption, ROX). The absolute ROX value was subtracted from the values of measured respiration.

## Statistics

Data are presented as mean ± SEM. Comparisons were made by unpaired t-test, 1-way or 2-way ANOVA as appropriate, followed by Tukey post hoc analysis. $p < 0.05$ was considered significant.

## Acknowledgements

Supported by British Heart Foundation grants PG/12/26/29477 (MZ/AMS), RG/13/11/30384 (AMS) and CH/1999001/11735 (AMS); and by grants from the Norwegian Health Association (6412 to E Aasum/ADH), the UK Medical Research Council (MC_U105663142 to MPM). MPM is a Wellcome Trust Investigator (110159/Z/15/Z).

## Additional information

### Competing interests

Michael P Murphy: Has a financial interest in and is on the scientific advisory board of Antipodean Pharmaceuticals, Inc which is commercialising MitoQ. The other authors declare that no competing interests exist.

### Funding

| Funder | Grant reference number | Author |
| --- | --- | --- |
| Norwegian Health Association | 6412 | Anne D Hafstad |
| Medical Research Council | MC_U105663142 | Michael P Murphy |
| Wellcome | 110159/Z/ 15/Z | Michael P Murphy |

| British Heart Foundation | PG/12/26/29477 | Ajay M Shah<br>Min Zhang |
| British Heart Foundation | RG/13/11/30384 | Ajay M Shah |
| British Heart Foundation | CH/1999001/1173 | Ajay M Shah |

The funders had no role in study design, data collection and interpretation, or the decision to submit the work for publication.

## Author contributions

Matthew Hancock, Anne D Hafstad, Data curation, Formal analysis, Validation, Investigation, Methodology; Adam A Nabeebaccus, Data curation, Investigation, Methodology; Norman Catibog, Synne S Hansen, Johanna Lanner, Angela Logan, Ioannis Smyrnias, Investigation, Methodology; Katrin Schröder, Resources; Michael P Murphy, Resources, Funding acquisition, Methodology; Ajay M Shah, Conceptualization, Supervision, Funding acquisition, Writing—review and editing; Min Zhang, Conceptualization, Data curation, Formal analysis, Supervision, Funding aquisition, Investigation, Writing—original draft, Project administration, Writing—reviewing and editing

## Author ORCIDs

Katrin Schröder (iD) http://orcid.org/0000-0002-3099-526X
Michael P Murphy (iD) https://orcid.org/0000-0003-1115-9618
Ajay M Shah (iD) http://orcid.org/0000-0002-6547-0631
Min Zhang (iD) https://orcid.org/0000-0003-1657-1337

## Ethics

Animal experimentation: The studies were complied in strict accordance with the UK Home Office Guidance on the Operation of the Animals (Scientific Procedures) Act, 1986 and King's College London institutional guidelines. All of the animal work was performed under the project licence number: 70/8889.

## Decision letter and Author response

Decision letter https://doi.org/10.7554/eLife.41044.019
Author response https://doi.org/10.7554/eLife.41044.020

## Additional files

### Supplementary files

• Transparent reporting form
DOI: https://doi.org/10.7554/eLife.41044.017

### Data availability

All data generated or analysed during this study are included in the manuscript and supporting files.

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
