## [Decision Letter]

Thank you for submitting your article "Myocardial NADPH oxidase-4 regulates the physiological response to acute exercise" for consideration by *eLife*. Your article has been reviewed by three peer reviewers, including Milica Radisic as the Reviewing Editor and Reviewer #1, and the evaluation has been overseen by Mark McCarthy as the Senior Editor. The following individual involved in review of your submission has agreed to reveal their identity: Kathy Griendling (Reviewer #2).

The reviewers have discussed the reviews with one another and the Reviewing Editor has drafted this decision to help you prepare a revised submission.

Summary:

The reviewers found that your work was novel and thorough.

Essential revisions:

1) Please comment on the severity of exercise studied.

2) Studies of mitochondrial respiratory capacity were undertaken on cardiac mitochondria isolated after exercise. Due to the time required for isolation, do the changes reported merely reflect damaged mitochondria?

3) Please consider an important comment by reviewer 3: "A lack of change in cardiac Nox2 content (Figure 1—figure supplement 1) is cited as evidence for a lack of involvement of this protein in exercise responses, but Nox2 activity is regulated (differently to Nox4) by assembly and activation of the NADPH oxidase complex rather than protein content. Can this protein therefore be excluded on the basis of Nox2 content? "

*Reviewer #1:*

Hancock et al. present a thorough study, performed in a murine model, to support an important role of Nox4 and Nrf2 in cardioprotection during exercise. This paper is specifically focused on delineating mechanisms that lead to the protective (and non-damaging) effects of exercise to the heart. Specifically, oxidative stress is increased during exercise, due to the higher metabolic requirements. This creates ROS overload that the heart is capable of managing. This paper demonstrates through the use of mostly Western blotting and functional measurements in transgenic animals that Nox4 regulated exercise performance by activating Nrf2, which in turn activates mitochondrial antioxidants. Overall, I find this study to be thorough and I have several questions that could further improve the work:

1) It would be useful to compare mitochondria density and structure in WT and csNrf2KO by TEM to rule out the possibility of other mitochondrial defects.

2) Figure 2—figure supplement 1, shows rather small differences in cardiac function at peak exercise. Although this might show up as significant in a statistical test, I wonder if it is of any biological relevance.

3) It would be good to see the entire Western blots in the supplement. Lanes are good in the main text.

Reviewer #2:

In this manuscript, the authors convincingly demonstrate that cardiac Nox4 mediates the cardiac response to exercise. Up until now, the role of Nox4 in the heart has been probed only in pathological conditions; this is the first study to show a physiological role for Nox4 in the heart, as the authors correctly point out. The study is state-of-the-art, especially with respect to the animal models and the fact that mechanism was investigated in vivo. In general, the data support the conclusions; however, a better characterization of the new csNrf2KO mice is needed before they can truly claim cardiomyocyte specificity.

Major concerns:

1) Why do the csNrf2KO look like heterozygotes? Both RNA and protein are only reduced by 50%? If this is because samples were taken from whole heart and include non-cardiomyocytes, please be specific. In addition, some confirmation that Nrf levels are not affected in other tissues is necessary before you conclude cardiomyocyte specificity.

2) The only unconvincing data are those shown in Figure 4B. While it is clear that exercise induces a decrease in H_2_O_2_ production, the slight decrease in baseline in csNrf2KO and the reported levels after exercise do not appear to be different from exercise in WT mice. So perhaps H_2_O_2_ levels are already low in csNrf2KO mice and exercise doesn't reduce them further. Please either provide statistics showing that these values are in fact different from WT-exercise or soften this conclusion.

Reviewer #3:

This paper provides data indicating that cardiac NADPH oxidase-4 (Nox4) is an essential regulator of exercise performance in mice. The authors provide evidence that myocardial Nox4 levels increase during acute exercise and trigger activation of the transcription factor Nrf2, with the induction of multiple endogenous antioxidants. Furthermore cardiomyocyte-specific Nox4-deficient (csNox4KO) mice or Nrf2-deficient (csNrf2KO) mice exhibit similar compromised exercise capacity with mitochondrial and cardiac dysfunction. They hypothesise that Nox4/Nrf2 axis therefore drive a hormetic response that is required for optimal cardiac mitochondrial and contractile function during physiological exercise. While the data are generally compelling there are number of issues with the paper:

1) Severity of exercise studied. The relevance of the data to physiological responses is dependent upon the nature of the exercise protocol for mice, which the authors describe as acute, moderate aerobic exercise. It is difficult to know how this description is justified. For habitually sedentary caged mice, the protocol described appears to be a severely incremental exercise regimen. Some consideration of this is required.

2) Studies of mitochondrial respiratory capacity were undertaken on cardiac mitochondria isolated after exercise. Due to the time required for isolation, what is the relevance of such measurements to the state of the mitochondria during exercise – do the changes reported merely reflect damaged mitochondria?

3) The authors hypothesise that increased Nox4 activity during exercise stimulates Nrf2 activation, but does the lack of any effect of MitoQ supplementation (a mitochondrial antioxidant) on the responses of control mice argue against this? Do the authors think this is related to the specificity of the action of MitoQ, or to different sub-cellular localisation of the Nox4 and MitoQ, or some other explanation?

4) A lack of change in cardiac Nox2 content (Figure 1—figure supplement 1) is cited as evidence for a lack of involvement of this protein in exercise responses, but Nox2 activity is regulated (differently to Nox4) by assembly and activation of the NADPH oxidase complex rather than protein content. Can this protein therefore be excluded on the basis of Nox2 content?

---

## [Author Response]

Essential revisions:1) Please comment on the severity of exercise studied.

The exercise regime that we used in this study is a moderate intensity exercise protocol, as reported previously (e.g. Kemi et al., J Appl Physiol 2002, Hafstad et al., 2011, Muthusamy et al., 2012). The co-author Anne Dr Hafstad and colleagues have extensive experience of exercise-intensity controlled protocols in mice on a C57Bl6 background (e.g. Hafstad et al., 2013; Hafstad et al., 2011). High intensity exercise protocols in C57Bl6 mice are typically conducted at a higher inclination (25%) and higher running speeds, and the mice can only sustain high intensity running for a few minutes. Continuous running for ~100min (> 1500m) is only possible with moderate intensity protocols. We have now clarified that this was a moderate intensity protocol in the Materials and methods (subsection “Acute exercise regimes”), Results (paragraph one) and Figure legends.

2) Studies of mitochondrial respiratory capacity were undertaken on cardiac mitochondria isolated after exercise. Due to the time required for isolation, do the changes reported merely reflect damaged mitochondria?

Mitochondrial preparations were isolated using the same methodology in all groups (KO and controls) and we allowed a 40 min rest period before experiments, so the differences are unlikely to be related to the isolation protocol or time. There were no differences among groups in the mitochondrial yield (protein concentration) and we know from previous work in the lab that respiratory function is stable for at least 2-3 hours after isolation (Hafstad et al., 2011). Furthermore, we formally tested for mitochondrial membrane integrity by examining the response to cytochrome C – which showed no significant differences among groups. These data are now shown in Figure 3—figure supplement 3 and referred to in the Results (subsection “Mitochondrial function is impaired in csNrf2KO mice after exercise”).

3) Please consider an important comment by reviewer 3: "A lack of change in cardiac Nox2 content (Figure 1—figure supplement 1) is cited as evidence for a lack of involvement of this protein in exercise responses, but Nox2 activity is regulated (differently to Nox4) by assembly and activation of the NADPH oxidase complex rather than protein content. Can this protein therefore be excluded on the basis of Nox2 content?"

We fully agree with the reviewer’s comment that Nox2 activation requires assembly of the oxidase complex and cannot therefore be inferred merely from Nox2 levels. We have now performed additional experiments in which we quantified the protein levels of the essential subunit p47^phox^ in the membrane fraction. Translocation of p47^phox^ to the membrane is a key step in Nox2 activation and is therefore a good surrogate for complex assembly and oxidase activation. The results showed no differences in membrane levels of p47^phox^ with exercise, indicating that there was no significant activation of Nox2. These new results are now included in Figure 1—figure supplement 2B and described in the Results (paragraph 1).

Reviewer #1:Hancock et al. present a thorough study, performed in a murine model, to support an important role of Nox4 and Nrf2 in cardioprotection during exercise. This paper is specifically focused on delineating mechanisms that lead to the protective (and non-damaging) effects of exercise to the heart. Specifically, oxidative stress is increased during exercise, due to the higher metabolic requirements. This creates ROS overload that the heart is capable of managing. This paper demonstrates through the use of mostly Western blotting and functional measurements in transgenic animals that Nox4 regulated exercise performance by activating Nrf2, which in turn activates mitochondrial antioxidants. Overall, I find this study to be thorough and I have several questions that could further improve the work:1) It would be useful to compare mitochondria density and structure in WT and csNrf2KO by TEM to rule out the possibility of other mitochondrial defects.

Guided by the editorial summary, we have not pursued this experiment in the current paper. However, please note that we now show data from experiments with cytochrome C addition to assess mitochondrial membrane integrity (Figure 3—figure supplement 3), which suggest that mitochondrial integrity is preserved.

2) Figure 2—figure supplement 1, shows rather small differences in cardiac function at peak exercise. Although this might show up as significant in a statistical test, I wonder if it is of any biological relevance.

We believe that the differences are indeed physiologically relevant. In awake echocardiography, we found a 24% increase in LV fractional shortening (FS) in the control group at peak exercise whereas csNox4KO mice actually showed a 6% decrease (Figure 2C). Given the high heart rates (>600 beats/min), this translates into a substantial change in cardiac output. The physiological relevance of this difference in cardiac function in cardiac-specific KO mice is also evident in the marked differences in exercise duration (our intervention in this experiment is cardiac-specific). We have now emphasised the significant differences in cardiac fractional shortening response in the Discussion (paragraph two).

3) It would be good to see the entire Western blots in the supplement. Lanes are good in the main text.

We have provided full images of all Western blots in Supplementary figures.

Reviewer #2:

*In this manuscript, the authors convincingly demonstrate that cardiac Nox4 mediates the cardiac response to exercise. Up until now, the role of Nox4 in the heart has been probed only in pathological conditions; this is the first study to show a physiological role for Nox4 in the heart, as the authors correctly point out. The study is state-of-the-art, especially with respect to the animal models and the fact that mechanism was investigated* in vivo*. In general, the data support the conclusions; however, a better characterization of the new csNrf2KO mice is needed before they can truly claim cardiomyocyte specificity.*

Major concerns:1) Why do the csNrf2KO look like heterozygotes? Both RNA and protein are only reduced by 50%? If this is because samples were taken from whole heart and include non-cardiomyocytes, please be specific. In addition, some confirmation that Nrf levels are not affected in other tissues is necessary before you conclude cardiomyocyte specificity.

As suggested by the reviewer, the mRNA and protein levels were in whole heart and the residual presence of Nrf2 is likely to mainly reflect expression in non-myocytes. We have now specifically commented on this point (subsection “Cardiomyocyte Nrf2 is required for optimal increments in heart performance during exercise”). We also checked Nrf2 mRNA levels in other organs to confirm cardiac-specific expression and now show these results in Figure 3—figure supplement 1.

2) The only unconvincing data are those shown in Figure 4B. While it is clear that exercise induces a decrease in H_2_O_2_ production, the slight decrease in baseline in csNrf2KO and the reported levels after exercise do not appear to be different from exercise in WT mice. So perhaps H_2_O_2_ levels are already low in csNrf2KO mice and exercise doesn't reduce them further. Please either provide statistics showing that these values are in fact different from WT-exercise or soften this conclusion.

The reviewer is correct that exercise induces a decrease in H_2_O_2_ levels in control mice whereas no such decrease occurs in csNrf2KO mice – consistent with the increase in endogenous antioxidants in control mice but no change in the KO mice. Indeed, the main point that we wished to highlight here was the different response to exercise rather than the absolute levels in each group. We have now re-phrased this sentence to make this clearer:

“We found in normal mouse hearts that mitochondrial H_2_O_2_ levels were significantly lower immediately upon cessation of peak exercise than in the non-exercising state (Figure 4B) – consistent with a reduction in ROS at peak exercise due to an enhanced antioxidant state. In marked contrast, there was no decrease in mitochondrial H_2_O_2_ levels with exercise in csNrf2KO mice, in line with the failure to enhance mitochondrial antioxidants.”

Reviewer #3:[…] 1) Severity of exercise studied. The relevance of the data to physiological responses is dependent upon the nature of the exercise protocol for mice, which the authors describe as acute, moderate aerobic exercise. It is difficult to know how this description is justified. For habitually sedentary caged mice, the protocol described appears to be a severely incremental exercise regimen. Some consideration of this is required.

Please see the response in Essential Revision above.

2) Studies of mitochondrial respiratory capacity were undertaken on cardiac mitochondria isolated after exercise. Due to the time required for isolation, what is the relevance of such measurements to the state of the mitochondria during exercise – do the changes reported merely reflect damaged mitochondria?

Please see the response in Essential Revision above.

3) The authors hypothesise that increased Nox4 activity during exercise stimulates Nrf2 activation, but does the lack of any effect of MitoQ supplementation (a mitochondrial antioxidant) on the responses of control mice argue against this? Do the authors think this is related to the specificity of the action of MitoQ, or to different sub-cellular localisation of the Nox4 and MitoQ, or some other explanation?

Based on our previous studies and those of others, we believe that Nox4 is located in the endoplasmic reticulum (e.g. Santos et al., 2016). Since MitoQ targets to the mitochondria, it is not surprising that it should have no effect on Nox4 actions in a different location. From the perspective of the overall exercise response and the effect of interventions on mitochondrial ROS, our results suggest that there is limited scope for enhancement of the adaptive exercise response in healthy adult mice with normal Nox4/Nrf2 activation, at least in the relatively acute exercise setting. However, this may be different in settings where Nox4/Nrf2 activation is impaired. We have discussed this in the penultimate paragraph of the Discussion section).

4) A lack of change in cardiac Nox2 content (Figure 1—figure supplement 1) is cited as evidence for a lack of involvement of this protein in exercise responses, but Nox2 activity is regulated (differently to Nox4) by assembly and activation of the NADPH oxidase complex rather than protein content. Can this protein therefore be excluded on the basis of Nox2 content?

We agree and we therefore performed additional experiments to address this point. Please see the response in Essential Revision above.